# The Dominant factors and optimization plan of soccer substitutions in Chinese Football Association Super League under the Five-substitution rule

Tong Chen, Liang Chen 📧*, Rong Li, KeHao Lv

School of Physical Education And Sport Science, Fujian Normal University, Fuzhou, Fujian, China

* cullencl@126.com

## Abstract

Substitution strategy is a fundamental component of tactical adjustments in soccer matches. Since the official introduction of the five-substitution rule in 2020, substitute players have faced increased performance demands. This study examines 2,125 substitution events (excluding goalkeepers) from the 2023 Chinese Football Association Super League (CSL) season, utilizing non-parametric tests (Kruskal-Wallis and Mann-Whitney) to compare soccer player performance differences under different substitution effectiveness conditions. A random forest model is also developed to assess the influence of contextual and player performance variables on substitution effectiveness, aiming to identify optimal decision-making pathways. Results show that, regardless of effectiveness, substitute players display significantly greater high-intensity running distances and passing accuracy than those they replaced. Effective substitutes also demonstrate superior ball recovery. Ineffective substitutions tend to occur earlier in the match, whereas effective and neutral substitutions are more concentrated in later phases. Effective substitutes perform better in medium-to-high-intensity running, passing accuracy, and shooting. Contextual variables contribute more to predicting substitution effectiveness, and the decision pathways suggest prioritizing players with high passing accuracy when leading, while targeted adjustments are needed when trailing.

## Introduction

Substitutions are a vital tactical tool in soccer, enabling coaches to influence the game within a restricted number of opportunities [1]. In 1994, FIFA introduced the "2+1" substitution rule, permitting an additional substitution if the goalkeeper was injured. In 1995, this rule was revised to allow three substitutions for any position. Due to fixture congestion during the COVID-19 pandemic, FIFA expanded the substitution limit to five per match in 2020. The Chinese Football Association Super League (CSL) also adopted this five-substitution rule. Analyses of European professional

**Data availability statement:** Data cannot be shared publicly because of third-party restrictions. Data are available from Opta Institutional Data Access (https://www.statsperform.com/opta/) for researchers who meet the criteria for access to confidential data. The data underlying the results presented in the study are available from Opta via email (lyt1721322454@outlook.com). Due to the third-party restrictions imposed by the official data provider, Opta, all personally identifiable information, including player names and club affiliations, has been anonymized. This anonymization ensures that the data are free from any intellectual property concerns while still preserving the integrity of the statistical analyses and outcomes.

**Funding:** This work was supported by National Social Science Foundation of China [23BTY044] and Fujian Provincial Social Science Planning Project [FJ2024B095].

**Competing interests:** The authors have declared that no competing interests exist.

leagues show that the expanded substitution rule led to a 48% increase in both average and total substitutions per match, reducing players' workload by 46% and enhancing coaches' tactical flexibility [2].

Well-executed in-game substitutions can shift attacking or defensive momentum [3], improve team and individual fitness, reduce fatigue [4,5], enhance running performance [6], and increase chances of scoring [7–9] and winning [10]. Substitution strategies often involve replacing underperforming, fatigued, or yellow-carded players [11]. Additional decisions are shaped by contextual factors such as match score, team strength, and venue [3]. Specifically, teams that are trailing tend to make earlier first substitutions with an offensive intent [3,4,12–14], whereas leading teams more often opt for defensive changes [5,13]. In matches against stronger opponents, first substitutions occur later [3,4], against weaker teams, more offensive changes are made, while evenly matched contests prompt defensive adjustments [5]. In tournaments, first substitutions occur earlier in group stages than in knockout rounds [12,13]. Match venue also plays a role: both home and away teams show increased scoring probability after the first and second substitutions. However, for away teams, this probability decreases after the third substitution. However, this effect is not immediately apparent within the first three minutes after a substitution [15].

Although there is general consensus on substitution scenarios and strategies, coaches often rely on intuitive "bounded rationality" during in-game decisions [16]. Existing studies have identified basic substitution patterns and partially examined their impact on match progression and outcomes. However, they often neglect contextual variables and do not offer optimal decision-making pathways. Moreover, most previous findings were based on the "three-substitution" rule, raising questions about their applicability under the "five-substitution" framework. Given that machine learning has been proven effective in predicting substitution outcomes [17], this study systematically incorporates match context and player performance variables to identify optimal substitution decision pathways using a random forest model.

## Methods

### Research subjects

The study focuses on 2,125 substitution events (excluding goalkeepers) from 240 soccer matches involving 16 teams during the 2023 CSL season from April 15th to November 4th. The data is sourced from OPTA, the official data provider for the CSL, which offers reports on physical and technical performance. A total of 41 indicators across three dimensions—player running performance (10 variables), technical performance (26 variables), and contextual variables (5 variables)—were utilized in this study.

### Non-parametric tests

Descriptive statistics of player performance samples with different substitution outcomes revealed that they did not follow a normal distribution. Therefore, non-parametric tests were employed to explore the differences between substitution outcomes and player performance.

Kruskal-Wallis Test was used for comparing differences in performance among players substituted into the match across three different substitution outcomes. The substitution outcome was treated as the grouping variable, while the performance of substituted players was the test variable. The Bonferroni correction method was applied for post hoc multiple comparisons. Mann-Whitney Test was used to compare the performance differences between substituted and substituted-out players within each substitution outcome group. Players (substituted in vs. substituted out) were used as the grouping variable, and substitution outcome was the test variable. Both non-parametric tests were conducted using SPSS 27.0.1 software.

Due to variations in playing time, player technical and running performance data were normalized by dividing them by playing time to enable direct comparisons. Indicators containing a high proportion of zero values (e.g., goals scored) were assessed using median and quartile values, though average rank was ultimately chosen for clearer differentiation.

## Random forest model

A random forest classification model was developed using Python 3.7.8 and the scikit-learn library. The model works by partitioning the dataset into multiple decision trees (DT) and then aggregating them into an ensemble for improved predictive accuracy [18,19]. Data processing and analysis were performed using the pandas library, while matplotlib was used for visualizing DT and feature importance rankings. This approach enables accurate prediction of the potential impact of various match situation variables on substitution outcomes and ranks these features based on their contribution to predictive accuracy.

## Performance evaluation metrics

Model performance was assessed using a confusion matrix and receiver operating characteristic (ROC) curves:

Confusion Matrix: A standard metric for evaluating classification models, summarizing predictions in a matrix format. Based on this, four key statistics were calculated: accuracy, precision, recall, and F1-score, to evaluate the model's classification ability in predicting substitution outcomes.

ROC Curve & Area Under Curve (AUC): A commonly used measure of classifier performance, where a larger AUC value indicates better diagnostic capability. This study separately evaluates AUC scores for predicting effective, neutral, and ineffective substitution outcomes.

## Research variables

Soccer substitutions are primarily based on player performance and match context. The contextual variables incorporated in this study include match score at substitution, venue (home = 1, away = 2), substitution timing, and team strength. Team strength was categorized based on final league standings: top-level teams (rank 1–5), mid-level teams (rank 6–11), and lower-level teams (rank 12–16). Player performance characteristics include 36 indicators from running and technical reports, all of which were normalized on a per-minute basis. Given the large number of match situation variables, a feature selection process was applied before model training to enhance the efficiency of the random forest classifier. Feature reduction was performed using Z-score standardization, followed by single-factor ANOVA (P ≥ 0.10) and multicollinearity diagnostics (VIF ≥ 5) to remove redundant variables. Ultimately, 24 player performance variables and 5 contextual variables were retained.

The match situation variables included in the model were match score difference at the time of substitution, match venue (home = 1, away = 2), substitution timing, team strength (top-level = 1, mid-level = 2, lower-level = 3), and player performance (average running and technical metrics). The substitution outcome served as the prediction variable. Building upon Mohandas' [17] binary classification of substitution effectiveness, this study improved the classification system into three categories: effective (1), neutral (0), and ineffective (−1).

After removing incomplete records, the final dataset included: 1,938 samples classified as effective (945 substituted-in players, 993 substituted-out players), 635 samples classified as neutral (319 substituted-in players, 316 substituted-out players), and 1,574 samples classified as ineffective (772 substituted-in players, 802 substituted-out players). A comprehensive definition and classification of all variables used in this study are presented in Table 1.

## Results

### Performance differences of substituted players under different substitution effects in the CSL

**Performance differences between substituted and replaced players under effective, neutral OR ineffective substitution.** A Mann Whitney test was conducted to compare the running and technical performance of substituted and replaced players under effective, neutral, and ineffective substitution effects. The results are shown in Table 2.

In effective substitutions, substituted players exhibited significantly higher average running distance, medium-to-high-intensity running distance (medium-speed running, fast running, high-speed running, and sprints), and high-intensity running frequency (high-speed running and sprints). They showed significantly lower jogging distance and maximum speed compared to replaced players. In terms of technical performance, substituted players had significantly higher percentages of successful passes, forward passes, and ball recoveries. However, they had significantly lower ball possession (individual ball control and turnovers), total passes (passes and forward passes), key area entries (live crosses, successful crosses, third-zone entries, and key passes), and shooting-related metrics (shots, shots on target, and goals). Defensive metrics such as tackling (tackles, successful tackles, tackle success rate) and dueling (aerial duels, successful aerial duels, aerial duel success rate, ground duels, successful ground duels, ground duel success rate, total successful duels, and duel success rate) were also significantly lower.

**Table 1. Definition and classification of variables in this study.**

| Variables | | Definition and classification of variables |
|---|---|---|
| Performance | Physical | Total distance、Standing distance、Jogging distance、Low speed running distance、Medium speed running distance、High speed running distance、Sprint distance、High speed running count、Sprint count、Maximum speed |
| | Technical | Offense: Personal ball control、Passing、Passing accuracy、Forward passes、Forward pass accuracy、Possession loss、Active crosses、Successful crosses、Passes into the attacking third、Key passes、Shots、Shots on target、Goals |
| | | Defense: Gained possession、Tackles、Successful tackles、Tackle success rate、Aerial Challenge for the ball、Successful aerial Challenge for the ball、Aerial Challenge for the ball success rate、Ground Challenge for the ball、Successful ground Challenge for the ball、Ground Challenge for the ball success rate、Challenge for the ball、Total successful Challenge for the ball、Challenge for the ball success rate |
| match situation | Match location | Home＝1, Away＝0 |
| | Teams' level | top-level＝1, mid-level＝2, lower-level＝3 |
| | Score change | The change in goal difference from the time of substitution to the final score was used as the predictive variable. |
| | Substitution time | The timing of the substitution event was also recorded. |
| Substitution Effects | Effective | If the final goal difference improved compared to the goal difference at the time of substitution, or if a leading position was maintained, the substitution was considered effective (coded as 1) |
| | Neutral | If both the goal difference at substitution and the final score were 0, the effect was considered neutral (coded as 0) |
| | Ineffective | If the final goal difference worsened or a losing position remained unchanged, the substitution was considered ineffective (coded as −1) |

Standing＝0～0.2 m/s; Jogging＝0.2～2 m/s; Low speed running＝2～4 m/s; Medium speed running＝4～5.5 m/s; High speed running＝5.5～7 m/s; Sprint=>7 m/s.

Table 2. Performance Differences Between Substituted and Replaced Players Under Different Substitution Effects.

| Variables | | Effective | | | | Neutral | | | | Ineffective | | | |
|---|---|---|---|---|---|---|---|---|---|---|---|---|---|
| | | Substituted (MR) | Replaced (MR) | Z | P | Substituted (MR) | Replaced (MR) | Z | P | Substituted (MR) | Replaced (MR) | Z | P |
| Physical | Total distance | 1055.53 | 887.63 | −6.602 | <0.001 | – | – | – | – | – | – | – | – |
| | Standing distance | – | – | – | – | 349.37 | 286.33 | −4.329 | <0.001 | 823.55 | 752.80 | −3.087 | 0.002 |
| | Jogging distance | 929.28 | 1007.77 | −3.086 | 0.002 | 291.83 | 344.42 | −3.612 | <0.001 | 755.10 | 818.69 | −2.775 | 0.006 |
| | Low speed running distance | 997.75 | 942.61 | −2.168 | 0.030 | 297.25 | 338.94 | −2.863 | 0.004 | 756.78 | 817.07 | −2.631 | 0.009 |
| | Medium speed running distance | 1004.53 | 936.16 | −2.689 | 0.007 | – | – | – | – | 810.72 | 765.15 | −1.989 | 0.047 |
| | High speed running distance | 1042.64 | 899.89 | −5.613 | <0.001 | 338.03 | 297.78 | −2.764 | 0.006 | 844.61 | 732.53 | −4.891 | <0.001 |
| | Sprint distance | 1000.24 | 940.25 | −2.359 | 0.018 | 338.07 | 297.74 | −2.769 | 0.006 | 814.05 | 761.94 | −2.274 | 0.023 |
| | High speed running count | 1003.37 | 937.27 | −2.599 | 0.009 | – | – | – | – | 816.76 | 759.34 | −2.506 | 0.012 |
| | Sprint count | 999.82 | 940.64 | −2.327 | 0.020 | 336.74 | 299.08 | −2.587 | 0.010 | 815.22 | 760.82 | −2.374 | 0.018 |
| | Maximum speed | 863.70 | 1068.02 | −8.039 | <0.001 | 284.68 | 348.32 | −4.381 | <0.001 | 747.53 | 825.97 | −3.423 | 0.001 |
| Technical | Personal ball control | 927.90 | 1009.09 | −3.193 | 0.001 | – | – | – | – | – | – | – | – |
| | Passing | 876.72 | 1057.79 | −7.120 | <0.001 | – | – | – | – | – | – | – | – |
| | Passing accuracy | 1350.62 | 606.80 | −29.249 | <0.001 | 429.29 | 205.66 | −15.359 | <0.001 | 1064.69 | 520.68 | −23.738 | <0.001 |
| | Forward passes | 896.64 | 1038.84 | −5.593 | <0.001 | – | – | – | – | 751.10 | 822.54 | −3.118 | 0.002 |
| | Forward pass accuracy | 1119.86 | 826.41 | −11.566 | <0.001 | 344.87 | 290.88 | −3.739 | <0.001 | 881.75 | 696.78 | −8.098 | <0.001 |
| | Possession loss | 922.59 | 1014.14 | −3.600 | <0.001 | – | – | – | – | – | – | – | – |
| | Active crosses | 833.81 | 1098.63 | −11.637 | <0.001 | 298.71 | 337.48 | −2.998 | 0.003 | 756.02 | 817.80 | −3.035 | 0.002 |
| | Successful crosses | 911.40 | 1024.80 | −7.124 | <0.001 | 306.31 | 329.80 | −2.914 | 0.004 | 763.07 | 811.01 | −3.592 | <0.001 |
| | Passes into the attacking third | 929.59 | 1007.48 | −3.098 | 0.002 | 295.08 | 341.14 | −3.208 | 0.001 | – | – | – | – |
| | Key passes | 876.27 | 1058.22 | −8.134 | <0.001 | 300.89 | 335.28 | −2.827 | 0.005 | 743.76 | 829.61 | −4.441 | <0.001 |
| | Shots | 939.06 | 998.47 | −2.573 | 0.010 | 290.62 | 345.64 | −4.232 | <0.001 | – | – | – | – |

(Continued)

Table 2. (Continued)

| Variables | Effective | | | | Neutral | | | | Ineffective | | | |
|---|---|---|---|---|---|---|---|---|---|---|---|---|
| | Substituted (MR) | Replaced (MR) | Z | P | Substituted (MR) | Replaced (MR) | Z | P | Substituted (MR) | Replaced (MR) | Z | P |
| Shots on target | 924.46 | 1012.37 | −4.668 | <0.001 | 295.45 | 340.76 | −4.755 | <0.001 | 751.13 | 822.51 | −4.84 | <0.001 |
| Goals | 943.94 | 993.83 | −3.628 | <0.001 | 310.06 | 326.02 | −3.318 | 0.001 | 764.00 | 810.13 | −5.636 | <0.001 |
| Gained possession | 1001.29 | 939.25 | −2.441 | 0.015 | – | – | – | – | – | – | – | – |
| Tackles | 886.40 | 1048.59 | −6.594 | <0.001 | 290.70 | 345.56 | −3.937 | <0.001 | 722.80 | 849.78 | −5.731 | <0.001 |
| Successful tackles | 898.31 | 1037.25 | −5.885 | <0.001 | 288.36 | 347.92 | −4.472 | <0.001 | 735.90 | 837.17 | −4.795 | <0.001 |
| Tackle success rate | 926.20 | 1010.71 | −3.582 | <0.001 | 296.06 | 340.15 | −3.310 | 0.001 | 755.98 | 817.84 | −2.933 | 0.003 |
| Aerial Challenge for the ball | 921.26 | 1015.41 | −3.790 | <0.001 | – | – | – | – | 759.37 | 814.58 | −2.472 | 0.013 |
| Successful aerial Challenge for the ball | 901.02 | 1034.67 | −5.862 | <0.001 | – | – | – | – | 745.77 | 827.67 | −4.109 | <0.001 |
| Aerial Challenge for the ball success rate | 723.39 | 1203.71 | −24.103 | <0.001 | 248.99 | 387.67 | −12.607 | <0.001 | 623.01 | 945.84 | −18.676 | <0.001 |
| Ground Challenge for the ball | 894.27 | 1041.09 | −7.100 | <0.001 | 284.47 | 351.84 | −5.842 | <0.001 | 721.32 | 851.20 | −7.097 | <0.001 |
| Successful ground Challenge for the ball | 913.66 | 1022.64 | −6.331 | <0.001 | 293.12 | 343.12 | −5.028 | <0.001 | 739.07 | 834.12 | −6.028 | <0.001 |
| Ground Challenge for the ball success rate | 919.63 | 1016.96 | −5.693 | <0.001 | 293.57 | 342.66 | −4.937 | <0.001 | 742.64 | 830.69 | −5.621 | <0.001 |
| Total successful Challenge for the ball | 897.88 | 1037.66 | −5.838 | <0.001 | 300.29 | 335.88 | −2.607 | 0.009 | 732.76 | 840.19 | −5.073 | <0.001 |
| Challenge for the ball success rate | 929.75 | 1007.33 | −3.246 | 0.001 | – | – | – | – | 755.20 | 818.59 | −2.999 | 0.003 |

MR denotes Mean Rank. denotes Mean Rank;– indicates a non-significant variable.

In neutral substitutions, substituted players covered a significantly higher standing distance, high-intensity running distance (high-speed running and sprints), and sprint frequency, but showed significantly lower medium-to-low-intensity running distance (jogging and medium-speed running) and maximum speed. Technically, they had a significantly higher pass success rate and forward pass percentage, but lower key area entries, shooting metrics, defensive tackles, and dueling performance.

In ineffective substitutions, substituted players covered a significantly higher standing distance, medium-to-high-intensity running distance (fast running, high-speed running, and sprints), and high-intensity running frequency, but replaced players exhibited significantly higher jogging distance, medium-speed running distance, and maximum speed. In terms of technical performance, substituted players had significantly higher pass success rates and forward pass percentages, but lower forward passes, key area entries, shooting metrics, defensive tackles, and dueling performance.

## Performance differences of substituted players under different substitution effects

A Kruskal-Wallis test was conducted to analyze the running and technical performance differences of substituted players under effective, neutral, and ineffective substitution effects. The results are presented in Table 3.

Compared to effective substitutions, neutral and ineffective substitutions showed that significantly lower Low speed running distance, forward pass percentage, shots on target, goals but significantly higher standing distance. Additionally, neutral substitutions showed that significantly lower average running distance and total shots than effective substitutions. Ineffective substitutions exhibited that significantly lower sprint distance, sprint frequency, and overall pass success rate, but higher live crosses. Ineffective substitutions occurred significantly earlier than effective and neutral substitutions, with median and interquartile substitution times of 70.00 (59.00–80.00), 74.00 (60.00–82.00), and 68.00 (58.00–78.00) minutes, respectively.

## Prediction and decision pathway for substitution effects in the CSL

**Performance of the random forest classification model.** In this study, the model was trained and optimized using five scenarios and 24 player performance features, as previously described. The dataset was split into training data (80%) and test data (20%). A combination of grid search and five-fold cross-validation was applied to train, predict, and fine-tune the random forest classification model. The results are presented in Fig 1, Tables 4, and 5.

Table 3. Non-parametric test of substitution player performance under different substitution effects.

| Variables | | Mean Rank | | | H | P |
|---|---|---|---|---|---|---|
| | | Effective | Neutral | Ineffective | | |
| Substitution time | | 1034.28 | 1112.32 | 960.41[ab] | 16.400 | <0.001 |
| Physical | Total distance | 1059.73 | 943.26[a] | 999.12 | 10.713 | 0.005 |
| | Stand distance | 968.73 | 1093.98[a] | 1048.23[a] | 14.005 | <0.001 |
| | Low speed running distance | 1061.57 | 956.76[a] | 991.30[a] | 10.243 | 0.006 |
| | Sprint distance | 1046.07 | 1047.42 | 972.81[a] | 7.514 | 0.023 |
| | Sprint count | 1054.19 | 1037.41 | 967.00[a] | 9.746 | 0.008 |
| Technical | Passing accuracy | 1068.37 | 1037.94 | 949.42[a] | 17.811 | <0.001 |
| | Forward passing accuracy | 1066.75 | 968.49[a] | 980.10[a] | 12.274 | 0.002 |
| | Active crosses | 978.95 | 1035.37 | 1059.94[a] | 12.872 | 0.002 |
| | Shot | 1051.13 | 974.81[a] | 996.61 | 8.205 | 0.017 |
| | Shots on target | 1055.68 | 974.10[a] | 991.33[a] | 20.540 | <0.001 |
| | Goals | 1053.91 | 986.67[a] | 988.3[a] | 51.009 | <0.001 |

[a]denotes a statistically significant difference from the effective group, [b]denotes a statistically significant difference from the neutral group。 All pairwise comparisons were adjusted using the Bonferroni correction.

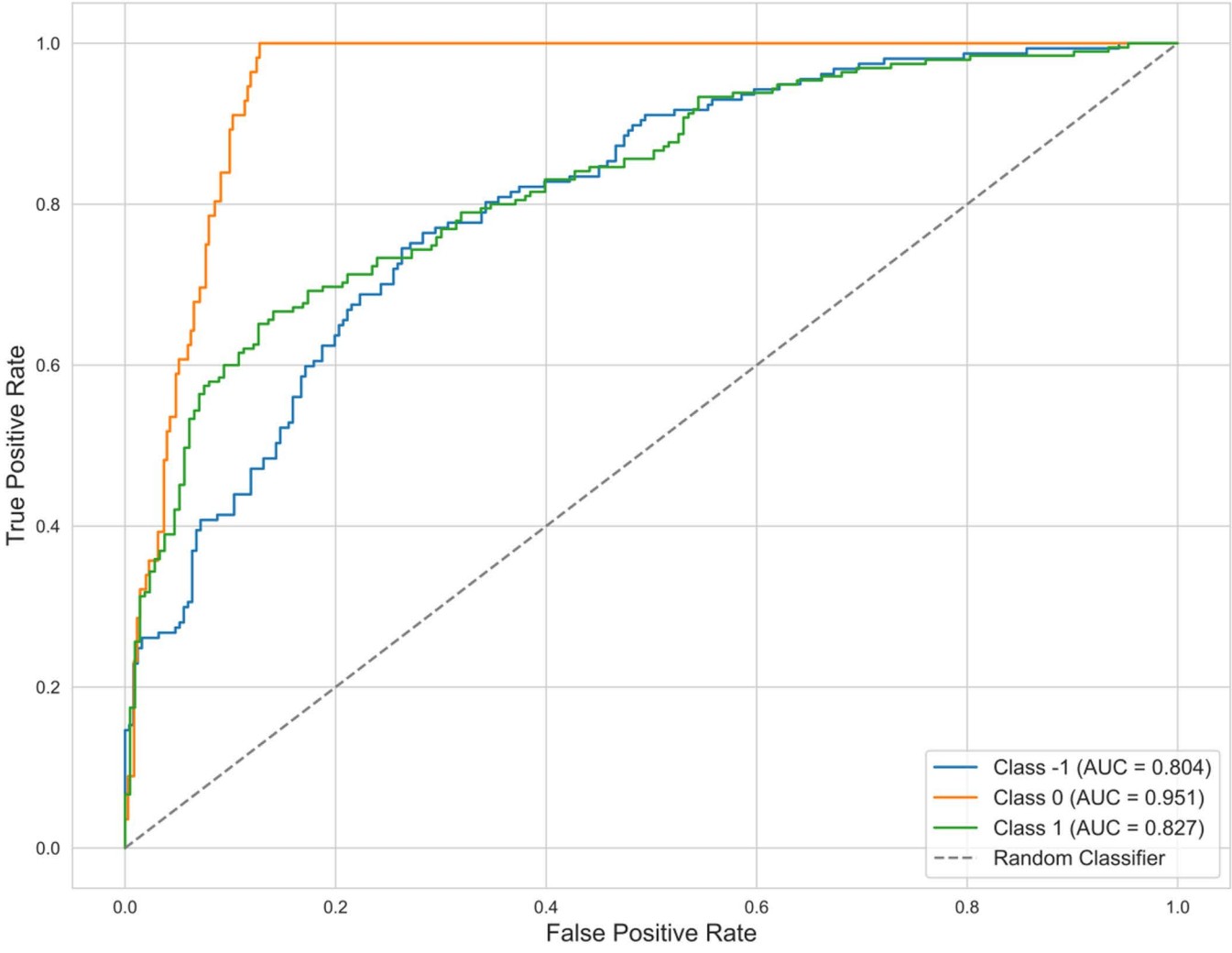

**Fig 1. ROC curves analysis of random forest model.**

As shown in Table 4, the model was constructed using 1,000 DT, and its optimal performance was achieved when the maximum depth was set to 4, yielding an AUC value of 0.861. Table 5 presents the model's performance in predicting the effectiveness of substituted players (effective, neutral, and ineffective) on the test set. The precision values for the three substitution effectiveness categories were 0.716, 0.633, and 0.652, while the recall values were 0.697, 0.679, and 0.656, respectively. The macro F1 scores were 0.706, 0.655, and 0.654, with an overall accuracy of 0.697, 0.679, and 0.656. After parameter tuning, the AUC values for predicting effective, neutral, and ineffective substitution outcomes on the test set reached 0.827, 0.951, and 0.804, respectively (see Fig 1).

### Core features in predicting substitution effects in the CSL

The study utilized the feature importance calculation method provided by scikit-learn to rank the key features contributing to the prediction of effective, neutral, and ineffective substitution outcomes based on the characteristics of the substituted players. The ranking results are illustrated in Fig 2.

**Table 4. Results of optimization of random forest model.**

| Model Parameters | Baseline Model | Optimal Model |
|---|---|---|
| n_estimators | 10 | 1000 |
| min_samples_split | 1 | 20 |
| min_samples_leaf | 1 | 4 |
| max_features | auto | sqrt |
| max_depth | None | None |
| bootstrap | FALSE | FALSE |
| random_state | 42 | 420 |
| auc | 0.835 | 0.861 |

**Table 5. Random forest model test set predicts performance.**

| Effectiveness | Precision | Recall | F1-Score | Accuracy |
|---|---|---|---|---|
| Effective | 0.716 | 0.697 | 0.706 | 0.697 |
| Neutral | 0.633 | 0.679 | 0.655 | 0.679 |
| Ineffective | 0.652 | 0.656 | 0.654 | 0.656 |

As shown in Fig 2, the match score status had the most significant impact on substitution outcome predictions. Apart from score status, the top five features contributing to effective substitution outcomes were opponent team strength, own team strength, passing success rate of the substituted player, standing distance, and forward pass success rate. For neutral substitution outcomes, the top five contributing features were all related to the substituted player's performance, namely standing distance, ball possession gains, high-speed running distance, fast running distance, and ball possession gains. The top five contributing features for predicting ineffective substitutions were opponent team strength, own team strength, passing success rate of the substituted player, standing distance, and fast running distance.

## Decision pathway of substitution effects prediction in the CSL

Through model construction and parameter tuning, it was found that the model performed best when the maximum depth was set to 4. Based on the feature selection and prediction results from 1,000 DT, the final decision pathway for substitution effectiveness prediction was established (see Fig 3).

As illustrated in Fig 3, when constructing the overall model for substituted players, match score status was set as the root node, with score status and own team strength as sub-nodes. This suggests that CSL teams primarily base their substitution decisions on the match score status. When the team is leading, if the team has strong overall strength, substituting a player with a high passing success rate is more likely to result in an effective substitution outcome. if the team has weaker overall strength, additional factors such as substitution timing and the player's jogging and fast running distances need to be considered: if the substitution occurs before the 68th minute, an effective substitution outcome is more likely when the substituted player has a shorter fast-running distance. if the substitution occurs after the 68th minute, an effective substitution outcome is more likely when the substituted player has a shorter jogging distance.

When the team is trailing, if the opponent team is strong, an effective substitution outcome is more likely if the substituted player scores a goal. If the opponent team is weak, an effective substitution outcome is more likely if the substituted player has a lower passing success rate. When the match is tied, the probability of achieving a neutral substitution outcome is generally higher for CSL teams.

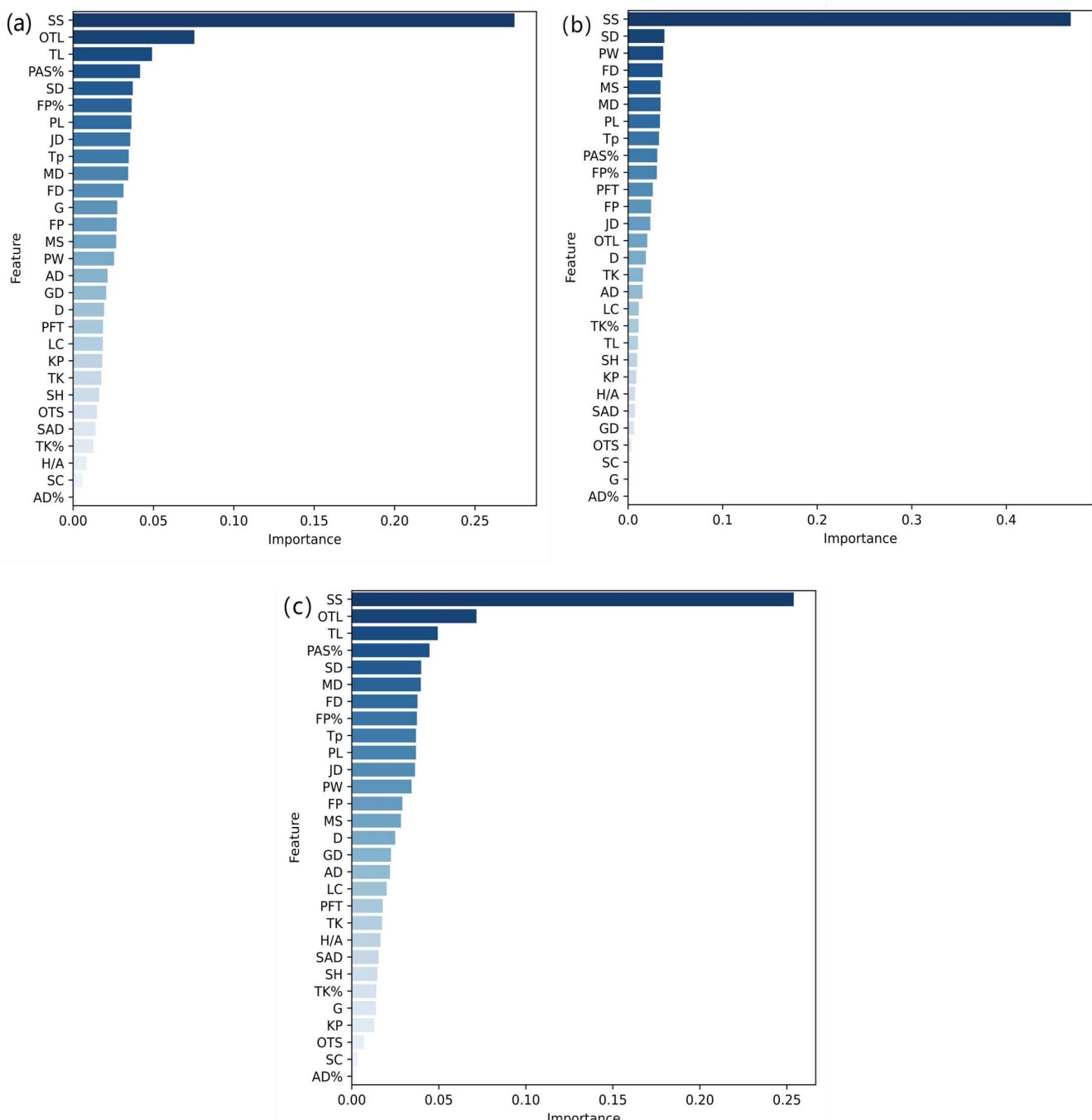

**Fig 2. Ranking of feature importance analysis of random forest model.** (a) Ranking of the importance of effective characteristics. (b) Ranking of the importance of neutral effect characteristics. (c) Ranking of the importance of ineffective characteristics. SS=Score Status; OTL=Opponent Team Levels; TL=Team Levels; PAS%=Passing Accuracy; SD=Stand Distance; FP%=Forward Passing Accuracy; PL=Possession Loss; JD=Jogging Distance; Tp=Time point; MD=Medium Speed Running Distance; FD=High Speed Running Distance; G=Goals; FP=Forward Passing; MS=Maximum Speed; PW=Gained Possession; AD=Aerial Challenge For the Ball. GD=Ground Challenge For the Ball; D=Challenge For the Ball Success rate. PFT=Passes into the Attacking Third. LC=Active Crosses. KP=Key Passes; TK=Tackles; SH=Shots; OTS=Shots on Target; SAD=Successful Aerial Challenge For the Ball; TK%=Tackles Success rate; H/A=Home/Away; SC=Successful Crosses; AD%=Aerial Challenge For the Ball Success rate.

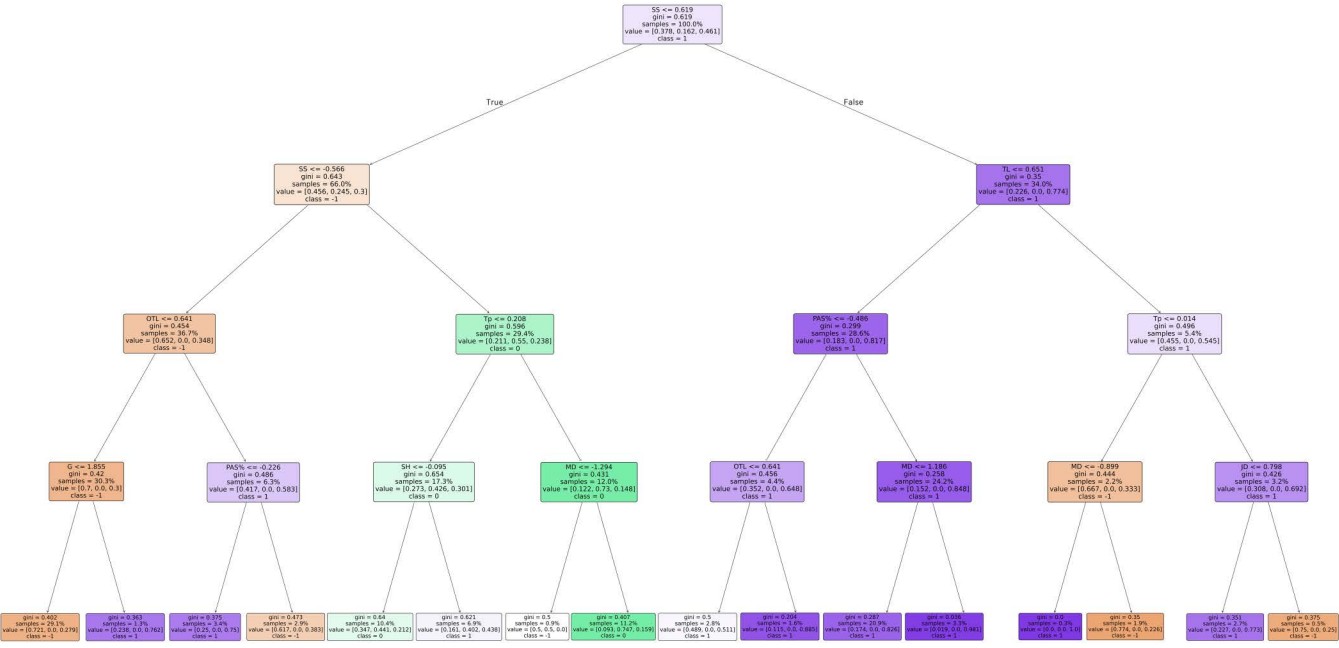

**Fig 3. Decision tree of effect prediction path for substitute players.**

## Discussion

This study quantifies the effect of substitutions (effective, neutral, and ineffective) based on changes in score and uses non-parametric tests to explore differences in performance between substituted players under these three effects. It also compares performance between substituted and substituting players across different substitution effects. To reflect the decision-making sequence of coaches, the study integrates comprehensive technical and physical performance indicators from OPTA reports, along with contextual variables—including match venue, opponent strength, score status, and substitution timing—to construct multiple decision pathways for varying substitution effects.

### Performance differences between substituted and substituting players under effective, neutral OR ineffective substitution in the CSL

High-intensity running—including sprints and high-speed movements—is a key performance indicator in modern soccer and plays a critical role in determining match outcomes [20]. This study finds that, regardless of whether the substitution effect is effective, neutral, or ineffective, substituted players consistently demonstrate superior high-intensity running, lower jogging distances, and slower maximum speeds compared to replaced players. Moreover, players substituted under neutral or ineffective conditions tend to cover more standing distance. On one hand, playing time is a major factor affecting match intensity [21], and most substitutions, whether under the 3-substitution or 5-substitution rule, occur in the midfield or the second half [22]. Thus, substituted players with shorter playing time are expected to sustain higher levels of high-intensity running. Prior studies under the three-substitution rule similarly found that incoming players generally outperformed outgoing players in high-intensity running [11,23–26]. Hence, it is concluded that, regardless of substitution effect or substitution rule, players substituted on tend to outperform substituted-off players in terms of high-intensity running.

While replacing a player with superior physical performance can influence match outcomes [6], low-intensity running does not create good scoring opportunities [27]. Therefore, players substituted on with more standing distance may not help the team change the score. Maximum speeds are typically reached during decisive moments—such as regaining

possession or outpacing opponents—which may determine match outcomes [28]. Bradley et al. (2014) found that, under the three-substitution rule, players taken off exhibited higher maximum speeds than substitutes [11]. This study also finds that, regardless of substitution effect, players substituted on exhibit significantly lower maximum speeds compared to those substituted off, thus highlighting the poorer leg strength and intermittent exercise capacity of substituted-on players [29].

Technically, substituted-on players under all substitution effects demonstrate higher passing and forward passing success rates than those they replaced. Additionally, only those substituted under effective outcomes show better ball recovery, while substituted-off players perform better in other attacking and defensive metrics. A prior study on the 2014 CSL reported that substitutes attempted more passes, shots, crosses, and duels than substituted-off or full-time players, but with lower success rates in passing, forward passing, and duels [26]. This study, based on the 2023 season after changes to substitution rules, shows that under the 5-substitution rule, players substituted on in the CSL have superior passing and forward passing success rates compared to players substituted off. This could be attributed to the rule change, which led to earlier substitution timings [12], giving substituted-on players more time to adapt and perform.

Possession reflects a team's control, organization, and offensive capability, and is strongly correlated with winning probability [30]. In the CSL, teams that fall behind tend to improve their possession to control the rhythm of the match and attempt to change the score [31]. The study finds that, while players substituted on under all three effects exhibit better passing success rates than those substituted off, players under the effective effect additionally show superior defensive performance in ball recovery. Combined with higher passing success, substituted players are more likely to help maintain leads or influence scoreline.

## Performance differences of substituted players under different substitution effects in the CSL

To identify characteristics associated with effective substitution outcomes, this study further analyzes performance differences among substituted players. Significant differences in performance and substitution timing were found among substituted players under different effects. Regarding timing, effective and neutral substitutions occurred significantly later than ineffective ones. A study of five major European leagues over five seasons under the three-substitution rule found that the optimal substitution timing for the best effect was between 60–75 minutes, with ineffective substitutions increasing linearly after 60 minutes [17]. In comparison, under the five-substitution rule in the CSL, effective and neutral substitutions tend to occur later, while ineffective ones cluster earlier. The expansion to five substitutions has led to earlier average substitution timing [12], and, combined with increased match intensity [22,32], has significantly improved substitute player performance.

Performance differences are also evident in running metrics. Players substituted on under effective effects tend to show better average running distance, mid-to-high intensity running, but shorter standing distances. Soccer players' workloads increase progressively throughout the match, leading to physical performance decline [33], especially in the second half when running performance decreases over time [34,35]. Liu et al. (2020) found that players substituted on later in the match exhibit superior mid-to-high intensity running [36]. Technically, players substituted under effective outcomes show higher passing accuracy and better shooting performance, but attempt fewer crosses. Passing accuracy is critical to team success, as it maintains possession [37], creates scoring chances, and limits the opponent's time on the ball [38]. Therefore, effective substitution is characterized by later substitutions with higher passing success rates and better shooting performance. In summary, making late substitutions by bringing on players who exhibit higher levels of moderate-to-high intensity running, better passing accuracy, and superior shooting performance—while replacing players with lower passing success rates and poor ball recovery ability—is more likely to help maintain a lead or improve the scoreline.

## Random forest prediction of substitution effects in the CSL

This study applied the random forest algorithm to analyze substitution effects in the 2023 CSL. Model performance metrics—accuracy, recall, precision, and macro F1—were consistent with prior machine-learning studies on substitution

prediction [39]. The model achieved AUC values of 0.827, 0.951, and 0.804 for effective, neutral, and ineffective substitutions, respectively, confirming its effectiveness in analyzing substitution strategies in CSL. To explore how substitution effects can be better predicted based on contextual and player performance variables, DT were generated to visualize the decision-making process of coaches in determining substitution effects.

Match score significantly influenced substitution outcomes, with contextual variables—such as score status, team strength, and substitution timing—appearing near the root nodes of the DT. This suggests that, in the CSL, situational variables take precedence over player performance when making substitution decisions. Similar findings have been observed in previous studies on the English Premier League and the five major European leagues, where contextual variables (such as first goal and opponent strength) played a decisive role in determining match outcomes [39,40].

The study revealed that when leading, substitution decisions should consider the team's own strength; when trailing, the opponent's strength becomes more critical; and when the score is tied, substitutions are less likely to influence outcomes. In specific decision-making paths, a team leading and possessing stronger strength is more likely to achieve effective substitution outcomes by substituting players with higher passing success rates. Prior studies have shown that coaches leading in score often opt for defensive substitutions [13], leveraging team strength and passing accuracy to retain possession and preserve the lead. Moreover, for weaker teams leading in the score, substituting players with superior high-intensity running could help maintain the advantage. To maintain a score advantage, weaker teams may need to engage in more passive defensive running. Most key defensive technical actions, such as tackles and interceptions, tend to occur during high-intensity activities [41]. Moreover, studies have shown that low- to moderate-intensity running may have a negative impact on match outcomes [27]. Thus, for weaker teams in the lead, substituting players excelling in high-intensity running—rather than low-intensity profiles—may be more effective in preserving the score advantage.

Although the majority of goals are scored by starting players, substitute players still contribute to 13.2% of the total goals, with the 5-substitution rule slightly increasing this from 12.5% to 15.9% [12]. The study also finds that substituting players to score goals is more likely to improve match outcomes when trailing against stronger opponents. When trailing against stronger teams, passive defense becomes more common, making proactive offensive substitutions more beneficial. This is consistent with Iglesias et al.'s (2022) study, which found that goals scored 5–10 minutes after substitutions are positively correlated with match victories [10]. Although the average substitution time has shifted earlier under the five-substitution rule, it still typically occurs in the latter part of the second half (69.2±14.6 minutes) [12], a period during which goals can have a decisive impact on match outcomes. Furthermore, the present study reveals that when trailing against weaker opponents, substitutions involving players with lower passing accuracy are more likely to help narrow the score gap. This may be explained by coaches favoring offensive substitutions against weaker teams to apply greater attacking pressure [5]. Offensive positions generally require players to attempt riskier passes; therefore, bringing on attacking players who are more willing to take such risks when trailing against weaker opponents may be advantageous in altering the course of the match.

When the score is tied, this study finds that substitutions are unlikely to change the score in the CSL. Similar findings have been observed in studies of World Cup and UEFA Champions League substitutions, where no clear substitution characteristics were found when the match was tied. In soccer matches, the influence of a tied score depends on the match stage and opponent strength. For weaker teams, a draw may be a good result in the group stage, which makes substitution characteristics complex when the score is tied.

## Conclusion

In conclusion, This research utilizing non-parametric tests to compare soccer player performance differences under different substitution effectiveness conditions. Additionally, a random forest model is constructed to identify the optimal decision-making pathways for effective substitutions. The results indicate that, regardless of substitution effectiveness, substitute players exhibit significantly higher high-intensity running distances and passing accuracy than substituted

players, with those in the effective substitution group demonstrating superior ball recovery ability. Ineffective substitutions tend to occur earlier in the match, whereas effective and neutral substitutions are more concentrated in later phases. Effective substitutes perform better in medium-to-high-intensity running, passing accuracy, and shooting. Contextual variables contribute more to predicting substitution effectiveness, and the decision pathways suggest prioritizing players with high passing accuracy when leading, while targeted adjustments are needed when trailing.

The findings of this study are valuable for professional soccer clubs and coaches providing a machine learning method into optimizing substitution strategies for improving team performance. Despite providing valuable insights into substitution strategies in CSL teams, the analysis is based on data from a single season, which may limit the generalizability of the findings.

## Supporting information

**S1 Table. Definition and classification of variables in this study.** A comprehensive definition and classification of all variables used in this study are presented in S1. The data underlying the findings of this study have been anonymized and are available in the supplementary files. These include the team substitution information of 2023 season CSL teams, including Match location, Teams' level, Score change, Substitution time, substituted and replaced players' performance. Due to the copyright restrictions imposed by the official data provider, Opta, all personally identifiable information, including player names and club affiliations, has been anonymized. This anonymization ensures that the data are free from any intellectual property concerns while still preserving the integrity of the statistical analyses and outcomes.
(XLSX)

**S2 Table. Performance Differences Between Substituted and Replaced Players Under Different Substitution Effects.** The S2 Table is primarily used to compare the differences in technical and physical performance between players in the positive, neutral and negative effect groups.
(XLSX)

**S3 Table. Non-parametric test of substitution player performance under different substitution effects.** The S3 Table is primarily used to compare the differences in technique and physical performance between the positive, neutral and negative effect groups.
(XLSX)

**S4-S5 Table.** Results of optimization of random forest model *AND* Random forest model test set predicts performance. The S4-S5 Table is raw data from the random forest method and includes player fitness and technical information as well as game situation information.
(XLSX)

**S1-S3 Fig. ROC curves analysis of random forest model, Ranking of feature importance analysis of random forest model *AND* Decision tree of effect prediction path for substitute players.** The S1-S3 Fig, like the S4-S5 tabular data above, are the raw data generated in the random forest method.
(XLSX)

## Author contributions

**Data curation:** Tong Chen.

**Formal analysis:** Liang Chen.

**Funding acquisition:** Liang Chen, Rong Li.

**Methodology:** Tong Chen, Liang Chen.

**Project administration:** Tong Chen.

**Validation:** Rong Li, Kehao Lv.

**Visualization:** Rong Li, Kehao Lv.

**Writing – original draft:** Tong Chen, Rong Li, Kehao Lv.

**Writing – review & editing:** Liang Chen.

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
