## [Decision Letter · Decision Letter 0]

Jun 27 2025

Dear Dr. Chen,

Thank you for submitting your manuscript to PLOS ONE. After careful consideration, we feel that it has merit but does not fully meet PLOS ONE’s publication criteria as it currently stands. Therefore, we invite you to submit a revised version of the manuscript that addresses the points raised during the review process.

We look forward to receiving your revised manuscript.

Kind regards,

Julio Alejandro Henriques Castro da Costa

Academic Editor

PLOS ONE

Journal Requirements:

This work was supported by National Social Science Foundation of China [23BTY044] and Fujian Provincial Social Science Planning Project [FJ2024B095].

4. In the online submission form, you indicated that data cannot be shared publicly because of copyright. Data are available from the OPTA Institutional Data Access for researchers who meet the criteria for access to confidential data.The data underlying the results presented in the study are available from cullencl@126.com

Reviewers' comments:

Reviewer's Responses to Questions

**Comments to the Author**

1. Is the manuscript technically sound, and do the data support the conclusions?

Reviewer #1: Yes

Reviewer #2: Yes

2. Has the statistical analysis been performed appropriately and rigorously?

Reviewer #1: Yes

Reviewer #2: I Don't Know

3. Have the authors made all data underlying the findings in their manuscript fully available?

Reviewer #1: Yes

Reviewer #2: Yes

4. Is the manuscript presented in an intelligible fashion and written in standard English?

Reviewer #1: No

Reviewer #2: Yes

Reviewer #1: It is to be commended that the authors have produced a study on football performance analysis that is both contemporary and exhaustive. The study is generally well-designed, employing suitable methods and presenting findings that contribute to the existing literature. The employment of a data-driven approach, coupled with the utilisation of contemporary machine learning techniques such as random forest, serves to enhance the scientific value of the study. It is submitted that the article was perused with great pleasure.

However, minor structural and linguistic improvements are suggested that should be considered prior to acceptance for publication.

As demonstrated in Table 2, the current form of the table is rather dense and challenging for readers to comprehend. The presentation of a large number of variables and test results together has been shown to reduce the readability of the table. Consequently, the table should be restructured to be more straightforward and lucid, with emphasis placed exclusively on statistically significant and interpretation-critical variables.

The Conclusion section is largely a reiteration of the Discussion section with regard to content. The conclusion should summarise the main findings of the study in a concise, original and striking way; it should also include possible application areas of the research or suggestions for future studies. It is submitted that reorganisation of the present section would contribute to the overall integrity of the study.

The employment of language can, on occasion, be somewhat lacking in fluency, with the result that ambiguity may be introduced into technical expressions. It has been observed that certain sentences appear to exert a translation effect. It is imperative that the language of the article be revised in order to achieve a more academic and natural English style. This process should be undertaken by an expert whose native language is English.

In conclusion, the study contains original contributions and, subject to the correction of the aforementioned points, will be suitable for publication.

Reviewer #2: Well done on the impactful findings in this study. The recommendations and limitations provided offer valuable insights and serve as useful references for enhancing performance strategies. However, there are several areas that could be improved to support better reader comprehension:

1.Please consider refining the subheadings within the Results and Discussion section to improve clarity and logical flow.

2.Ensure that all abbreviations are introduced with their full terms upon first mention to facilitate understanding.

3. Review the overall formatting for consistency and alignment with academic presentation standards.

**Do you want your identity to be public for this peer review?** For information about this choice, including consent withdrawal, please see our Privacy Policy

Reviewer #1: **Yes: ** Görkem Açar

Reviewer #2: No

---

## [Author Response · Author response to Decision Letter 1]

21 May 2025

Thank you for giving us the opportunity to submit a revised draft of our manuscript titled "The Dominant factors and optimization plan of soccer substitutions in Chinese Football Association Super League under the Five-substitution rule" [PONE-D-25-18355] to PLOS ONE. We appreciate the time and effort that you and the reviewers have dedicated to providing valuable feedback on our manuscript. We have incorporated changes that reflect all the suggestions provided by the reviewers. All changes are highlighted in red in the revised manuscript. Please see below for our point-by-point responses to the reviewers' comments. We hope that our work can be improved again.

Response to Comments of Reviewer #1

Comment 1�As demonstrated in Table 2, the current form of the table is rather dense and challenging for readers to comprehend. The presentation of a large number of variables and test results together has been shown to reduce the readability of the table. Consequently, the table should be restructured to be more straightforward and lucid, with emphasis placed exclusively on statistically significant and interpretation-critical variables.

Response to comment 1: We appreciate this thoughtful suggestion. Upon review of Table 2, we acknowledge that the presentation of numerous variables could reduce its clarity. As you pointed out, one variable lacked statistical significance, and we have therefore removed it from the table.

The revised Table 2 now exclusively includes those variables that demonstrate statistical significance. In addition, we only described statistically significant variables in the results section. This change enhances the clarity of both the table and the overall discussion of the results.

Comment 2�The Conclusion section is largely a reiteration of the Discussion section. The conclusion should summarise the main findings of the study in a concise, original and striking way; it should also include possible application areas or suggestions for future studies.

Response to comment 2: We agree with your observation and have rewritten the Conclusion section to present the core findings in a more concise and original manner, highlight the practical implications of the study for football performance analysis, and provide possible application areas based on our results.

Comment 3:The employment of language can, on occasion, be somewhat lacking in fluency, with the result that ambiguity may be introduced into technical expressions. It has been observed that certain sentences appear to exert a translation effect. It is imperative that the language of the article be revised in order to achieve a more academic and natural English style. This process should be undertaken by an expert whose native language is English.

Response to comment 3: Thank you for pointing this out. To address this, we have had the entire manuscript professionally edited by a native English-speaking professor. The revised version reflects improvements in fluency, clarity, and academic tone. We trust this resolves the language issues previously identified.

Response to Comments of Reviewer #2:

Comment 1: Please consider refining the subheadings within the Results and Discussion section to improve clarity and logical flow.

Response to comment 1: We appreciate this suggestion. We have revised the subheadings throughout the Results and Discussion section to better reflect the thematic structure and logical progression of our analysis. This improves navigation and enhances the reader’s understanding.

Comment 2: Ensure that all abbreviations are introduced with their full terms upon first mention to facilitate understanding.

Response to comment 2: We carefully reviewed the entire manuscript and ensured that all abbreviations are now introduced with their full terms upon first use to assist the reader.

Comment 3:Review the overall formatting for consistency and alignment with academic presentation standards.

Response to comment 3: We conducted a thorough formatting review of the manuscript. The revised version aligns with the journal’s style guide in terms of headings, references, table and figure formatting, and general presentation standards. We have also re-checked the layout for consistency.

Think you and best regards.

Yours sincerely

CHEN Tong

Corresponding author:

Name: CHEN Liang

E-mail: cullencl@126.com

---

## [Decision Letter · Decision Letter 1]

The Dominant factors and optimization plan of soccer substitutions in Chinese Football Association Super League under the Five-substitution rule

PONE-D-25-18355R1

Dear Dr. Chen,

We’re pleased to inform you that your manuscript has been judged scientifically suitable for publication and will be formally accepted for publication once it meets all outstanding technical requirements.

Kind regards,

Julio Alejandro Henriques Castro da Costa

Academic Editor

PLOS ONE

Additional Editor Comments (optional):

Reviewers' comments:

Reviewer's Responses to Questions

**Comments to the Author**

Reviewer #1: All comments have been addressed

Reviewer #2: All comments have been addressed

2. Is the manuscript technically sound, and do the data support the conclusions?

Reviewer #1: Yes

Reviewer #2: Yes

3. Has the statistical analysis been performed appropriately and rigorously?

Reviewer #1: Yes

Reviewer #2: I Don't Know

4. Have the authors made all data underlying the findings in their manuscript fully available?

Reviewer #1: Yes

Reviewer #2: Yes

5. Is the manuscript presented in an intelligible fashion and written in standard English?

Reviewer #1: Yes

Reviewer #2: Yes

Reviewer #1: Dear Editor,

First of all, I would like to congratulate the researchers for focusing on a highly current, original and application-oriented topic. I believe that such scientific studies in the field of sports science, especially in high-performance sports such as football, will be well received both in academic literature and in field applications.

The authors have carefully reviewed all the criticism and suggestions we have provided, demonstrating a constructive attitude and making the necessary corrections with great diligence. The explanations provided throughout the revision process are detailed, clear, and convincing. This clearly demonstrates the authors' understanding of scientific ethics, their openness to criticism, and their mastery of the field.

The study is methodologically sound; the sample selection, data collection tools, and analysis techniques are appropriately structured. The findings are presented in a statistically meaningful manner, and the results are discussed in a manner consistent with the literature. Furthermore, the study will clearly serve as a guide for field practitioners, coaches, and performance analysts.

In conclusion, due to the high scientific contribution and practical value of the study, I believe that the revised final version of this article is suitable for publication.

Reviewer #2: (No Response)

**Do you want your identity to be public for this peer review?** For information about this choice, including consent withdrawal, please see our Privacy Policy

Reviewer #1: **Yes: ** Görkem Açar

Reviewer #2: No

---

## [Editor Report · Acceptance letter]

PONE-D-25-18355R1

PLOS ONE

Dear Dr. Chen,

I'm pleased to inform you that your manuscript has been deemed suitable for publication in PLOS ONE. Congratulations! Your manuscript is now being handed over to our production team.

Kind regards,

on behalf of

Dr. Julio Alejandro Henriques Castro da Costa

Academic Editor

PLOS ONE